Force-velocity-power variables derived from isometric and dynamic testing: metrics reliability and the relationship with jump performance

http://orcid.org/0000-0002-6027-4367 Vieira Amilton 1 amiltonvieira@unb.br
http://orcid.org/0000-0002-1865-2790 Cunha Rafael 1
Gonçalves Carlos 1
Dal Pupo Juliano 2
http://orcid.org/0000-0001-8325-0344 Tufano James 3
1 University of Brasília, Faculty of Physical Education, Brasília , Distrito Federal , Brazil
2 Biomechanics Laboratory, Center of Sports, Federal University of Santa Catarina , Florianópolis, SC , Brazil
3 Faculty of Physical Education and Sport, Charles University in Prague , Prague , Czech Republic
Jimenez Manuel
Electronic publication date: 2024 Nov 8
Publication date: 2024
Volume: 12
Electronic Location ID: e18371
Received 2024 Apr 24; Accepted 2024 Sep 30
Copyright: © 2024 Vieira et al.
Copyright year: 2024
Copyright holder: Vieira et al.
License: This is an open access article distributed under the terms of the Creative Commons Attribution License, which permits unrestricted use, distribution, reproduction and adaptation in any medium and for any purpose provided that it is properly attributed. For attribution, the original author(s), title, publication source (PeerJ) and either DOI or URL of the article must be cited.
License URL: https://creativecommons.org/licenses/by/4.0/

Keywords: Assessment, Performance, Dynamic strength index, Two-point method, Linear regression, Isometric midthigh pull, Squat jump

Funding: Fundação de Apoio à Pesquisa do Distrito Federal (FAPDF) Edital DPI/DPG/BCE 01/2024 from the University of Brasilia This work was funded by the Fundação de Apoio à Pesquisa do Distrito Federal (FAPDF). The APC was funded by Edital DPI/DPG/BCE 01/2024 from the University of Brasilia. The funders had no role in study design, data collection and analysis, decision to publish, or preparation of the manuscript.

==============================
We investigated the convergent validity and intrasession reliability of force, velocity, and power (FVP) variables and the dynamic strength index (DSI) obtained from isometric midthigh pull (IMTP) and squat jump (SJ) testing. Fifteen male combat sports athletes (27 ± 5 years, 77 ± 9 kg, 1.76 ± 0.1 m, 14 ± 6% body fat) participated in a 2-days study. The first day involved testing familiarization, while the second was dedicated to IMTP and SJ testing. Maximal isometric force (Fiso) was obtained from IMTP, while mean force, mean velocity, jump height, and jump impulse (J) were gathered from SJ. To analyze the FVP, we calculated the linear relationship between force and velocity, which allowed us to obtain the slope of the relationship (SFV), the theoretical velocity at zero force (V0), and the theoretical maximal power (Pmax). DSI was obtained as a ratio from SJ peak force and Fiso. The convergent validity was investigated using Spearman’s ρ coefficients to assess the relationships between jump height and J with Fiso, V0, SFV, Pmax, and DSI. The intrasession reliability was assessed using intraclass correlation coefficients (ICC) and coefficient of variations (CV). All variables demonstrated acceptable reliability scores. ICC ranged from moderate to excellent, and the mean CV was <10%. We found a “very large” correlation between jump J and Pmax, while jump height was not correlated with any variable. In conclusion, the IMTP and SJ combination is a practical way to determine FVP producing capacities that can be reliably measured (intrasession). The Pmax, derived from FVP, was correlated with jump performance, which might evidence the convergent validity of the method.

Introduction

Increasing muscle strength can improve health and athletic performance (Suchomel et al., 2018) while also reducing injury risk (Lauersen, Andersen & Andersen, 2018; Suchomel, Nimphius & Stone, 2016). In an effort to increase strength, many athletes perform strength training, but they also likely perform conditioning activities and sport-specific training. With so many factors at play, it is possible that training interference may occur, which could result in suboptimal adaptations. Throughout the years, many methods of athlete monitoring have taken shape to determine how an athlete is responding to training and whether or not certain adaptations are occurring. Among the many athlete monitoring methods, the dynamic strength index (DSI) is commonly used since it only requires two tests (one force-biased and the other velocity-biased) and theoretically describes the general strength profile of an athlete (Thomas, Jones & Comfort, 2015; Suchomel et al., 2020).

In theory, the DSI (also known as the dynamic strength deficit) is a diagnostic that indicates whether an athlete lacks maximum force production, lacks explosive force production, or is well-balanced between the two. As different athletes require different force attributes, there is no ideal DSI that all athletes should aim for, but it is instead used to help coaches or athletes identify primary targets (Comfort et al., 2018). To apply the DSI concept, the athlete must perform an isometric test (such as the isometric mid-thigh pull, IMTP) and an explosive dynamic test (such as the squat jump), with the ratio of peak force between the two formulating the DSI. A ratio <0.6 indicates that the athlete’s profile is more maximum-force dominant (and may want to focus on ballistic training), while a ratio >0.8 indicates a more ballistic athlete who may want to incorporate more heavy loads in training if they desire a balanced profile. Despite the general concern about the reliability of ratios or indexes data such as DSI (Bishop, Shrier & Jordan, 2023), previous studies suggest that DSI are highly reliable (Thomas, Jones & Comfort, 2015; García-Sánchez et al., 2024).

As half of the DSI equation includes force output during an explosive test like a vertical jump, it would be assumed that DSI is closely related to vertical jump height, which is commonly used to determine the power output of an athlete. However, controversy exists concerning the relationship between DSI and jump performance, with some studies showing no relationship between DSI and jump height (JH) (Secomb et al., 2015; Suchomel et al., 2020) and others demonstrating moderate to large correlations (Pleša et al., 2024). This controversy might be related to methodological issues. While Secomb et al. (2015) and Suchomel et al. (2020) investigated DSI obtained from IMTP performed around 135° knee angles, Pleša et al. (2024) revealed correlations when the participants were tested with knees flexed at 90° angle. Regardless of whether a relationship between DSI and JH exists, it is important to note that combining information from isometric and dynamic testing may be far more informative than only providing the DSI.

For example, in the strength and conditioning literature, force, velocity, and power (FVP) producing capacities seem to be useful to show a spectrum of performance abilities ranging from theoretical maximal force ( F0), velocity ( V0), power ( Pmax), and the slope of the force and velocity relationship ( SFV), which can bring information on muscles’ mechanical capacity and can guide training prescription (Jaric, 2016; Morin & Samozino, 2016). Though many methods of determining FVP exist, a two-point method, using the two most distinct loads, has been pointed out as a time-savvy way (Pérez-Castilla et al., 2018). However, it is important to note that all mechanical variables ( F0, V0, P0, and SFV) are extrapolated and consequently more sensitive to estimation error. One conceivable solution to this issue would be to avoid extrapolations by directly measuring force at null velocity and velocity at null force. While measuring force at null velocity is feasible by measuring the maximal isometric force ( Fiso), measuring velocity at null force is still challenging (Rivière et al., 2023). Although some concerns about comparing Fiso and F0 extrapolated from dynamic actions may exist (Rahmani et al., 2001), measuring the Fiso would be a partial solution, avoiding at least the F0 extrapolation.

Although testing Fiso and jump performance (i.e., two-point method) is likely a simple and practical way of determining the FVP producing capacities, there is still a need to validate the obtained variables against a performance measurement related to the neuromuscular function or sports performance. In this context, the JH would be an intuitive choice since it has been commonly used within athlete testing batteries, distinguishing athletes with higher from lower competitive levels and diverse training backgrounds (Cormie, McGuigan & Newton, 2010; James et al., 2020; McMahon, Lake & Comfort, 2022). Therefore, it would be expected that athletes who demonstrate “better” FVP capacities, notably Pmax, would jump higher than those who exhibit inferior FVP producing capacities (Cronin & Sleivert, 2005). However, it should be noted that JH is affected by body mass (it is harder for heavy individuals to jump high), so the jump take-off momentum or jump net impulse (J) might be more related to neuromuscular function. However, to our knowledge, no study investigates the relationship between jump J and FVP producing capacities or DSI. Data from athletes of other modalities may suggest that Jump J may be a promising metric for collision and combat sports since it reveals the mechanical aspect of the movement (driven and strategy determinants of the performance output), which might aid sports scientists in ranking or monitoring athletes’ performance (McMahon et al., 2020, McMahon, Lake & Comfort, 2022). Therefore, we aimed to investigate the convergent validity and the reliability of force, velocity, and power profiling variables obtained via a two-point method using the isometric midthigh pull and squat jump testing. We hypothesized that FVP variables would demonstrate acceptable reliability scores and that significant correlations would be found between FVP variables and DSI with jump J, but those metrics would not correlate with jump height.

Materials and Methods

Participants

A total of 15 male combat athletes (27 ± 5 years; 1.75 ± 0.1 m; 76 ± 9 kg; 14 ± 7 fat %) from different disciplines (e.g., judo, jiu-jitsu, karate, taekwondo, MMA) participated in the study. The sample size was established using GPower® software considering the potential “very large” correlation (H1 = 0.7; H0 = 0) between jump performance and force-velocity profiling metrics; two-tailed test with α of 0.05 and of power (1−β) 0.8. These assumptions indicated that a sample size of at least 13 athletes would reach sufficient power to avoid type II errors. The inclusion criteria required that participants were engaged in any modality of combat sport at least 3 days per week for at least 2 years. They had 14 ± 7 years of experience in their modality and were frequently exposed to strength and conditioning practices. As exclusion criteria, athletes currently performing rapid weight loss, suffering from any injuries that could compromise maximal testing performance, or not following our recommendation to avoid vigorous exercise 48 h before testing days were not allowed to participate. The participants were informed about the risks and benefits of the research. The study followed the Helsinki Declaration’s ethical standards, and the participants signed an informed consent form. The research was approved by the local Ethical Committee (number 3.796.898).

Study design

The participants were invited to perform a 2-days testing protocol, including familiarization and isometric mid-thigh pull (IMTP) and squat jump (SJ) testing (Fig. 1). The first day served as a familiarization where participants performed as many practice trials as needed, filled out the forms (e.g., training practices and routines), and underwent height and DEXA scans. The following day (2 to 7 days apart), they completed 5 min of a standardized warm-up protocol composed of dynamic stretching and body weight exercises, followed by 50%, 75%, and 90% of the perceived maximum IMTP trials, each for 5 s. After 3-min, participants performed three maximal IMTP for 5 s and SJ under the supervision of a single rater. All tests were performed on a 101 × 76 cm force plate (Accupower Portable Force Plate; AMTI, Watertown, MA, EUA). Each maximal effort was separated by 1-min of rest.

Figure 1 Study design.

Squat jump test

Using athletic shoes, participants were required to step over the force plate and then achieve a squat position with ∼90° of knee and hip flexion determined using a universal handheld goniometer. To maintain the vertical trunk position and to control the squat depth, the participants held a 0.5 kg PVC pipe in their upper back (near C7). They maintained the squat position for 2–3 s and then were verbally encouraged to jump as high as possible immediately following an auditory jump command (“three, two, one, jump”). The participants were motivated to jump higher by seeing the jump height displayed on a monitor. The force-time curve was inspected after each SJ, and if a counter movement occurred (force >5% of the body weight), the trial was repeated. Force-time data was processed as previously published (Ferreira et al., 2023). Briefly, body weight and mass were measured during the one second period of the weighing phase with a lower standard deviation (SD). The start of the SJ was identified as the instant when the force signal exceeded the threshold of 5 × SD. Velocity was calculated by numerically integrating acceleration-time using the trapezoidal rule. Jump take-off was identified when the force signal reached the threshold of 5 × SD of the flight force (platform unloaded). Jump net impulse (J) was calculated by multiplying take-off velocity and body mass following the impulse-momentum theorem.

Isometric midthigh pull test

The IMTP test was performed in a custom-made rack (Select Fit, Brasília, Brazil), with the participants standing on the force platform (Couto et al., 2023). The rack allowed for adjustments in the bar height with a precision of 1 mm. Participants were positioned with their feet approximately hip-width apart and hands approximately shoulder-width apart. The bar height was adjusted to correspond to the second pull position of the clean exercise. This procedure resulted in knee and hip angles of 140 ± 5° and 145 ± 10°, respectively (180° = full extension). Joint angles were measured using a universal handheld goniometer, while feet and hand distances were determined with an anthropometric measuring tape. These measurements were performed on the first day and repeated on the second. A single rater verbally encouraged the participants to produce a maximal effort in each attempt. The IMTP initiation was set as the time when the force rose to 5 × SD body weight. Pre-tension was controlled not to exceed 50 N, and data without a stable period of at least 1 s or presenting a counter movement immediately before the force rise or the maximal force only in the last second (4 s of the trial) were excluded.

Dynamic strength index and force-velocity-power profiling

The metrics were extracted from vertical ground reaction forces using a custom-made Python script based on previous recommendations (McMahon et al., 2018; Comfort et al., 2019). The signal was filtered using a fourth-order low-pass Butterworth filter with a cutoff frequency of 30 Hz. This low-pass threshold was determined based on previous studies successfully applying this cutoff frequency (Samozino et al., 2008; Vieira et al., 2023). DSI was calculated as the ratio of SJ peak force to IMTP peak force. For FVP profiling, IMTP peak force was considered Fiso, and velocity was assumed to be zero. We calculated the linear relationship between Fiso and jump mean force and velocity, which allowed us to obtain the slope of the relationship (SFV=−Fiso/V0), the theoretical velocity at zero force (V0=−Fiso/SFV), and Pmax as a function of Fiso×V0/4 (Jaric, 2016).

Statistical analyses

From three IMTP and three SJ tests, we selected the two greater values ( Fiso from IMTP and mean force and mean velocity from SJ) for reliability purposes, and the mean of both was used in the correlations analysis. The normality of the data was examined using the Shapiro-Wilk test. Fiso ( p = 0.01) and SFV ( p = 0.04) were not normally distributed. Intrasession test-retest was assessed using two-way mixed effects intraclass correlation coefficients (ICC) for absolute agreement (Koo & Li, 2016) and coefficient of variation (SD divided by the mean times 100). Spearman’s ρ coefficients were used to assess the relationships between JH and jump J with Fiso, V0, SFV, Pmax, and DSI. The magnitude of each relationship was interpreted as trivial (0.00–0.09), small (0.10–0.29), moderate (0.30–0.49), large (0.50–0.69), very large (0.70–0.89), and nearly perfect (0.90–1.00) (Hopkins et al., 2009). All statistical tests were performed using IBM SPSS (version 26, Armonk, NY, USA) with statistical significance set at 0.05.

Results

All variables demonstrated acceptable intrasession reliability scores (Fig. 2). ICC ranges from moderate to excellent, and the mean CV was <10%. We found no correlation between any variable with JH, but Pmax demonstrated a “very large” correlation with jump J (Fig. 3).

Figure 2 Intrasession reliability of force-velocity-power variables obtained during the isometric mid-thigh pull and squat jump testing.

Data are presented as mean with a 95% confidence interval (CI, vertical lines) of the coefficient of variations (upper panel) interpreted as good (<5%, green), moderate (5% to 10%, yellow), and poor (>10%, red), while the intraclass correlation coefficients with 95% CI were interpreted as moderate (0.50-0.749), good (0.75-0.90), and excellent (>0.90).

Figure 3 Correlation matrix between force-velocity-power variables and squat jump (SJ) height and jump net impulse (J).

The scatter plot depicting the “very large” relationship between net J and Pmax (two-tailed p = 0.001).

Discussion

We investigated the convergent validity and intrasession reliability of FVP variables and DSI obtained from two practical tests–IMTP and SJ. We found that the FVP variables and the DSI were reliable in athletes within a single session, and our investigation pointed out that Pmax presented a very large correlation with jump J, demonstrating evidence of convergent validity. These findings indicated that the FVP producing capacities can be easily determined by performing only IMTP and SJ tests and then applying the two-point method.

Our result demonstrating that DSI is reliable in combat athletes (ICC of 0.95 and CV of 1.9%) agrees with previous studies (Thomas, Jones & Comfort, 2015; García-Sánchez et al., 2024) reporting ICC values of 0.80 and 0.97, and CV of 4.6% and 6.1% in male college athletes (e.g., soccer, boxing, rugby) and semi-professional handball players, respectively. However, DSI showed no correlation with JH ( ρ = 0.09, p = 0.75, Fig. 3) in this study. Since DSI may be considered a as proxy measure for SFV (we found that these two were strongly correlated ρ = 0.94), the absence of correlation between DSI and jump performance was at least partially expected since it was suggested that SFV and jump performance might present inverse “U-shape” relationship, with SFV positively affecting performance until a certain level, passing this level the influence becomes negative (Samozino et al., 2014). Furthermore, others (Secomb et al., 2015; Suchomel et al., 2020) have directly reported the lack of correlation between DSI and jump performance. Secomb et al. (2015) found no correlation ( r = −0.20 to −0.32, p > 0.05) investigating the relationship between DSI obtained from IMTP and countermovement (CMJ) and SJ heights in adolescent surfing athletes, while Suchomel et al. (2020) found a similar result ( r = 0.11, p > 0.05) in a larger sample of 155 NCAA division I collegiate athletes. This lack of correlation between DSI and JH might also be due to JH being largely affected by body mass; it is harder for heavy individuals to jump high since greater body mass impedes the effective acceleration of the body mass, reaching greater takeoff velocity. However, a recent study found moderate ( r = 0.41) and large ( r = 0.63) correlations between DSI and, CMJ and SJ height, respectively (Pleša et al., 2024). More specifically, they found that isometric force (i.e., Fiso), and consequently DSI values, were primarily affected by body posture (150°, 120° and 90° knee angles), and the moderate and strong correlations were only observed when isometric force was measured at a 90° knee angle. While this result might suggest that isometric force should be measured at a 90° knee angle, a deeper analysis of their results points to the opposite (Pleša et al., 2024). It was noted that the mean jump force was equal to (DSI ratio = 0.99) or even greater (1.11) than the isometric peak force, which violates the well-established force-velocity inverse relationship. However, it is important to note that our testing protocol was designed to allow participants to produce their greatest isometric force ( Fiso), which is in line with the results of Pleša et al. (2024), who showed that peak force occurred at a 150° knee angle, which was similar to the 140 ± 5° angle used in our study (our study 3,095 ± 669 N vs. ∼3,000 N in their study). We showed a more realistic 0.84 ± 0.13 DSI value at these knee angles, corresponding to the 0.64 ± 0.19 DSI for the 150° knee angle in their study (Pleša et al., 2024). Therefore, the collective results of our study and others (Samozino et al., 2014; Secomb et al., 2015; Suchomel et al., 2020) demonstrated that jump performance and DSI are not correlated, leading coaches to question whether either should be used in isolation or if other metrics could better indicate whether an athlete lacks maximum force production, lacks explosive force production, or is well-balanced between the two. In this regard, determining the individual FVP producing capacities may provide a proxy measure of muscles’ mechanical capacity and can guide training prescription, as theoretically and experimentally demonstrated in previous studies (Samozino et al., 2014; Jaric, 2016; Morin & Samozino, 2016).

On the other hand, Šarabon, Kozinc & Marković (2020) argued against the use of an isometric-jumping approach to derive FVP variables because they found poor to fair validity scores for F0, V0, SFV, but not for Pmax, which they described as highly valid. The author’s argument was based on comparing the isometric-jumping method against the multiple-point method (assumed as a criterion). However, we noticed issues against using multiple overloaded jumps as a criterion method. First, it estimates F0 and their estimation was 32% (2,244 vs. 2,962 N) less than measured at 150° knee angle (the greatest force output measured). Second, the study was performed with a heterogeneous sample of participants, including athletes and recreationally active individuals from both sexes, who were familiarized and tested in a single session. These issues might have affected the overloaded jumps testing protocol, designed to begin from jumps with zero overload up to a jump height <7.5 cm using 10 kg increments. It can be noted that some individuals jumped against only three loads, while others jumped against 11 loads and then performed the isometric maximal testing. We assumed the fatigue level differed between participants, which may have compromised testing results. Furthermore, non-athletes might not produce reliable force data from overload jumps (Fessl, Wiesinger & Kröll, 2022). Therefore, considering the feasibility of the current protocol, which is less prone to induce fatigue, we are arguing in favor of the two-point method for directly measuring the force at null velocity (i.e., Fiso) instead of estimating it.

Although the present study provides exciting insights into determining the individual’s maximal neuromuscular capacity and testing for prescription, the approach is not free of limitations. Our results suggest that the FVP variables are reliable during a single testing session but must also be investigated between days to allow for athlete monitoring (i.e., 7 days apart or longer). Additionally, although we observed a “very large” correlation between Pmax and jump J, correlation does not mean causation. It must be investigated longitudinally, which is even more relevant because the correlation between jump J and Pmax was derived from the same vertical jump test, making the variables susceptible to covariance. Furthermore, although we have chosen a body posture that supposedly allowed the participants to produce their maximal force (Rahmani et al., 2001), it was not directly measured; it is well-known that the force output is angle-dependent. Furthermore, we are proposing a link between the maximal isometric force generated by the lower limbs at a specific body posture and the mean force produced over the range of the SJ motion. However, it should be noticed the link between isometric and dynamic force production warrants further research. Finally, the present results apply to male combat athletes, and further studies should investigate whether similar results might also be found in other populations, including athletes from other modalities or non-athletic populations.

As a practical application, the present results suggest that IMTP and SJ tests serve to evaluate the neuromuscular function in combat athletes. Combining results from both tests can provide the athlete and coach deeper insight into the mechanical capacities of the lower limbs by quantifying the FVP producing capacities, which is far more informative than using a single test outcome like IMTP force or SJ height or combing data from the two tests to obtain DSI. For example, the FVP profile allows one to distinguish “strong”, “fast”, and “powerful” individuals. In addition, the Pmax reveals the optimal external load maximizing power output since the maximum power produced is Fiso/2 at the velocity V0/2 (Jaric, 2016). Since athletes’ FVP can be quickly measured using the two-point method, it is less time-consuming than a traditional procedure requiring several loading conditions. Therefore, the current FVP approach may be easier to include in monitoring testing routines for muscle mechanical capacities or to evaluate adaptations that might occur in response to the athletes’ training program. Furthermore, we can suggest that all those metrics presenting a low coefficient of variation, i.e., the “green zone” shown in Fig. 2, are potential metrics to be utilized or tested in a longitudinal monitoring study.

Conclusions

The force-velocity-power variables are reliable intrasession, and the very large correlation between Pmax and jump impulse is evidence of the convergent validity of the two-point method using jump and isometric maximal tests. These promising results encourage further investigations applying the FVP variables obtained from isometric and jump tests for diagnosis and further investigating its effectiveness through a longitudinal study.

Supplemental Information

Supplemental Information 1 Raw data.

Additional Information and Declarations

Competing Interests

Author Contributions

Human Ethics

Data Availability

The authors declare that they have no competing interests.

Amilton Vieira conceived and designed the experiments, performed the experiments, analyzed the data, prepared figures and/or tables, authored or reviewed drafts of the article, and approved the final draft.

Rafael Cunha conceived and designed the experiments, performed the experiments, analyzed the data, authored or reviewed drafts of the article, and approved the final draft.

Carlos Gonçalves analyzed the data, authored or reviewed drafts of the article, and approved the final draft.

Juliano Dal Pupo conceived and designed the experiments, authored or reviewed drafts of the article, and approved the final draft.

James Tufano conceived and designed the experiments, authored or reviewed drafts of the article, and approved the final draft.

The following information was supplied relating to ethical approvals (i.e., approving body and any reference numbers):

The study was conducted in accordance with the Declaration of Helsinki for studies involving humans and The Research Ethics Committee (Comitê de Ética em Pesquisa-CEP-Plataforma Brasil) granted ethical approval to carry out the study (protocol number 3.796.898).

The following information was supplied regarding data availability:

The raw measurements are available as a Supplemental File.

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
