# Peer review of "Force-velocity-power variables derived from isometric and dynamic testing: metrics reliability and the relationship with jump performance"

_PeerJ, doi:10.7717/peerj.18371_

## Round 0.1 · original submission · Major Revisions

Dear authors:

Thank you for submitting your paper to PeerJ Journals. After peer review, we consider MAJOR REVIEWS. Please be sure to note the PDF of review comments from Rev 1

Please respond to all the following comments.

Regards

Dr. Manuel Jiménez

Reviewer 1 ·

Basic reporting

Authors need to adjust all citations in the text.

Experimental design

The authors need to provide more information in the introduction, methodology and discussion of the study

Validity of the findings

The authors need to make it clear that the data cannot be extrapolated to other populations.

Annotated reviews are not available for download in order to protect the identity of reviewers who chose to remain anonymous.

Reviewer 2 ·

Basic reporting

The tests that comprise the dynamic strength index must be better described in the introduction.
Line 44: citation of the dynamic strength index.

Experimental design

Line 76: Participants – Were there no exclusion criteria?
Lines 96 – 98: How long is the effort for the IMTP test? 5 seconds like warming up?
Line 102: “…holding a 0.5 kg PVC pipe before jumping.” – Describe better.

Validity of the findings

Make it clear in the discussion that the sample was composed of combat athletes and caution should be taken in possible extrapolations.

Additional comments

Great job. The paper reads well and makes a significant contribution to the literature.

Reviewer 3 ·

Basic reporting

I congratulate Authors and encourage such great initiative to better link research and practice. However, I suggest to Authors to include additional content and try to clarify their work, even though, based on the discussion section, the topic seems quite clear to their eyes.

Also, I stay open for further discussion, i.e., another round of review, in the case my comprehension of the work was wrong.

Authors should add a paragraph and scientific references on the relationship between F0 and maximal isometric force measurements, which have been address largely in the literature, aside from the dynamic strength index literature. I strongly believe that this paragraph would bring more useful content about the relationship between isometric measurements and FVP profiling, and help clarify the (conceptual) differences between the two types of measurement (dynamic versus isometric). I recommend this paper as a humble starting point to address this issue into the manuscript:
Rivière JR, Rossi J, Jimenez-Reyes P, Morin JB, Samozino P. Where does the One-Repetition Maximum Exist on the Force-Velocity Relationship in Squat? Int J Sports Med. 2017 Nov;38(13):1035-1043. doi: 10.1055/s-0043-116670. Epub 2017 Oct 1. PMID: 28965339.

English and some sentences need to be checked again.

Any other content are of enough quality for publication

Experimental design

The work may address an original aim, but the content is not enough clear and need clarification to highlight its originality.

Even though I understand the idea of the work (not the aim though) the terms “FVP profiling”, as well as derived variables (F0, V0, Sfv, Pmax) should not be used here, because the measurements and the methods are not representative of the usual force-velocity-power relationship assessment in squat jump.

Any other content are of enough quality for publication

Validity of the findings

The convergent validity addressed with the correlation between jump performance variables and other variables, maybe problematic because the tested correlations hide a non-negligible covariance. For example, with the correlation between the net impulse during the squat jump and Pmax. As Pmax is derived from SJ performance (that is, in detail, from V0 and the force developed during the squat jump), it shares variance with the net impulse, and thus are de facto correlated. The non-perfect correlation is in fact explained by the variance of “F0” (maximal isometric force), which is an independent measurement from the squat jump. Thus, I permit myself to question the convergent validity.

Any other content are of enough quality for publication

Additional comments

From a personal point of view:
I really ask myself whether measuring force and velocity during a SJ + a 1-RM (or a close load is enough, even though it is not the true 1RM) takes more time than the force (and velocity) measurement during a SJ + isometric mid thigh pull, notably regarding the time to set up for each individual the two stands of measurement. Based on my humble experience and knowledge of the literature, IMTP is quite hard to set up properly, RFD and Fmax are reliable only with very high quality measurements. Regarding authors results, I can only congratulate them !

I am not a fan of mixing dynamic (SJ) and isometric measurement (IMTP) in the same index, because, in my opinion they are “two different worlds”, to say it briefly; the literature is unclear and there is still a gap between, the two worlds. For example, establishing the link between a maximal force during IMTP at a specific angle and mean force over a full range of motion during SJ is not that easy to interpret.

Another thought is regarding the DSI, which could mimic Sfv, but only when F0 and IMTP max force are close to each other, which is not always the case. In addition, as DSI is somewhat a proxy for Sfv, it will be difficult anyway for it to be strongly correlated with performances, because Sfv influence only slightly performance comparing to Pmax, and when correlating Sfv with performance, Pmax is not fixed between athletes.

---

## Round 0.2 · accepted · Accept

Dear Authors:

It is a pleasure for us to inform you that your manuscript - Force-velocity-power variables derived from isometric and dynamic testing: metrics reliability and the relationship with jump performance - has been Accepted for publication.

Congratulations!

Dr, Manuel Jiménez

Reviewer 1 ·

Basic reporting

The authors put in a lot of effort. The authors responded and addressed all my suggestions in the introduction.

Experimental design

The authors responded and addressed all my suggestions in the methods.

Validity of the findings

The authors responded and addressed all my suggestions in the results and discussion.

Additional comments

Authors need to make a detailed analysis in English.

Reviewer 2 ·

Basic reporting

No comment

Experimental design

No comment

Validity of the findings

No comment

Additional comments

No comment